# Spherically Symmetric Configurations in Unimodular Gravity

Júlio César Fabris [1,2,*], Mahamadou Hamani Daouda [3] and Hermano Velten [4]

1 Núcleo Cosmo-Ufes & Departamento de Física, Universidade Federal do Espírito Santo, Av. Fernando Ferrari, 514, Goiabeiras, Vitória 29060-900, Brazil

2 Moscow Engineering Physics Institute, National Research Nuclear University MEPhI, Kashirskoe Shosse 31, Moscow 115409, Russia

3 Département de Physique, Université de Niamey, Niamey 10662, BP, Niger; daoudah77@gmail.com

4 Departamento de Física, Universidade Federal de Ouro Preto (UFOP), Campus Universitário Morro do Cruzeiro, Ouro Preto 35402-136, Brazil; hermano.velten@ufop.edu.br

* Correspondence: julio.fabris@cosmo-ufes.org

**Abstract:** Unimodular gravity (UG) is often deemed comparable to General Relativity (GR) in many respects, despite the theory exhibiting invariance under a more limited set of diffeomorphic transformations. The discussion we propose in this work relies on the criteria for establishing the equivalence between these two formulations, specifically exploring UG's application to static and spherically symmetric configurations with the energy-momentum tensor originating from either a scalar field or an electromagnetic field. We find that the equivalence between UG and GR might be disrupted when scrutinizing the stability of solutions at a perturbative level.

**Keywords:** unimodular gravity; black holes; stability





## 1. Introduction

General Relativity (GR) is the modern theory of gravitational interaction. The gravitational phenomena are considered as the structure of the space-time itself induced dynamically by matter. GR is considered a very successful theory: all local tests confirm the predictions of GR with high precision. At cosmological scales, it leads to the Standard Cosmological Model (SCM) which also addresses consistently all available observations, from scales of galaxies up to the larger structures of the universe. It accounts also successfully for the different phases of the evolution of the universe, including the primordial phases at least to the primordial nucleosynthesis scales. From this point of view, the SCM, based on GR, is an almost perfect model to describe the entire evolution of the universe.

However, seen from a different perspective, the SCM is at least problematic. To account for the observations at the different scales, it demands the introduction of two until now undetected components in the matter/energy content of the universe [1,2]. The dynamics of galaxies and cluster of galaxies, and even the formation of such structures, asks for an additional pressureless component, dubbed dark matter, which manifests only indirectly. Moreover, to explain the present accelerated phase of the universe, the CMB spectrum and to obtain an age of the universe consistent with the age of globular clusters, the SCM asks for another component, with negative pressure, which does not agglomerate, dubbed dark energy.

Dark energy is now frequently associated with vacuum energy as predicted by quantum field theory. However, its observed value seems not consistent with the theoretical predictions by dozens of orders of magnitude [3–5]. There are many proposals to cope with this problem. One of them is to replace it with a self-interacting scalar field, called quintessence [6]. However, it must be explained, in the quintessence program, why the vacuum energy must be exactly or, at least, nearly zero. Therefore, the vacuum energy must, somehow, *degravitate* [7–9]. There are many mechanisms to implement such *degravitating*

mechanism, but until now such proposals are, in some sense, in construction. For a general overview of the dark energy problem, see Ref. [10].

One interesting approach to the cosmological constant problem described above is through the unimodular gravity (UG) class of theories [11–14] where the determinant of the metric, $g$, is fixed. Originally, $g = 1$, but other possibilities can be explored, see next section. UG leads to traceless gravitational equations. The energy-momentum tensor does not conserve necessarily anymore, since UG is not invariant by the full diffeomorphism group, but by a more restricted structure called transverse diffeomorphism [15]: While the general diffeomorphism is characterized by a coordinate transformation $x^\mu \rightarrow x^\mu + \xi^\mu$, $x^\mu$ being an arbitrary vector, the transverse diffeomorphism requires that this function satisfies the condition $\xi^\mu_{;\mu} = 0$, as it will be seen later. If the conservation of the energy-momentum tensor is imposed, GR is recovered with a cosmological term that appears as an integration constant. If the conservation of the energy-momentum tensor is not imposed, as we will prove below, a class of *dynamical vacuum* theories is obtained, implying an interaction of the matter sector with the decaying cosmological term.

The present motivation for the UG deviates from the original one, which is closer to the choice of a coordinate system in GR. From the modern perspective, UG has two main motivations. The first one is related to the cosmological constant problem: if we introduce a cosmological constant term in the matter content, it disappears from the UG gravitational equations. It may reappear if the energy-momentum tensor is supposed to conserve, but the possible connection with the quantum vacuum energy density is lost and the usual cosmological constant problem is alleviated since it remains only to explain the observed value, but without a necessary connection with the vacuum energy. UG may have some interesting properties at a quantum level related to the volume fixing constraint, see Ref. [14] and references therein.

However, there are many discussions, from the original formulation of UG to its present applications, if the theory is equivalent to GR with a constant or dynamical cosmological term. In previous works we have explored the distinction between GR and UG mainly at a perturbative level in the cosmological context, see [16] and references therein (see also [17]). Here we will extend such studies to the static, spherically symmetric configurations. In UG, with the imposition of the conservation of the energy-momentum tensor, the static, spherically symmetric solutions are identical to those of GR but now contain a cosmological constant. The non-conservation of the energy-momentum tensor, on the other hand, can be mapped in the GR structure with a dynamical cosmological term. Indeed, the non-conservation of the energy-momentum tensor is allowed in this context, leading to a new formulation of the UG theory. It is also worth mentioning that such a non-conservation mechanism appears in many other situations. For a review, see Ref. [18]. All these aspects are discussed in the next section. In Section 3 the general equations for a static, spherically symmetric configuration are settled out. Some examples of interacting models, resulting from the non-conservation of the energy-momentum tensor, will be shown in Section 4, both in the presence of an electromagnetic field as well as of a self-interacting scalar field in Section 5. For the latter case, we perform, in Section 6, a perturbative analysis aiming to show how the usual results of GR change in the unimodular context. In particular, the unimodular condition on the determinant of the metric implies vanishing perturbations at the linear level. The results obtained are discussed in Section 7.

## 2. Field Equations

The Einstein–Hilbert action, in the presence of the cosmological term and the matter Lagrangian,

$$\mathcal{S} = \int d^4x \sqrt{-g} \left\{ \frac{R}{16\pi G} + 2\Lambda + \mathcal{L}_m \right\},\tag{1}$$

implies in the following field equations:

$$R_{\mu\nu} - \frac{1}{2}g_{\mu\nu}R = 8\pi G T_{\mu\nu} + g_{\mu\nu}\Lambda. \tag{2}$$

The application of the Bianchi identities leads to the energy-momentum tensor $T^{\mu\nu}$ conservation:

$$T^{\mu\nu}{}_{;\mu} = 0. \tag{3}$$

The conservation laws related to the energy-momentum tensor can be alternatively deduced from the invariance of the Einstein–Hilbert Lagrangian by diffeomorphic transformations [19].

In order to obtain the UG equations, we introduce a constraint in the action via a Lagrange Multiplier $\chi$ and an external field $\xi$ [17]:

$$\mathcal{S} = \int d^4x \left\{ \sqrt{-g}R - \chi(\sqrt{-g} - \xi) \right\} + \int d^4x \sqrt{-g}\mathcal{L}_m. \tag{4}$$

The presence of the external field allows one to use a suitable coordinate system according to the problem under analysis, for example, the usual spherical coordinates or the quasi-global coordinates employed in spherical symmetric space-time.

The final field equations for this case are

$$R_{\mu\nu} - \frac{1}{4}g_{\mu\nu}R = 8\pi G\left(T_{\mu\nu} - \frac{1}{4}g_{\mu\nu}T\right), \tag{5}$$

$$\frac{R^{;\nu}}{4} = 8\pi G\left(T^{\mu\nu}{}_{;\mu} - \frac{1}{4}T^{;\nu}\right). \tag{6}$$

The above Equation (6) is obtained by using the Bianchi identities in (5).

As highlighted in Ref. [19], it is important to note that in UG, the conservation of the energy-momentum tensor cannot be derived through the conventional diffeomorphism invariance because the theory only exhibits invariance with respect to a limited set of diffeomorphisms, referred to as transverse diffeomorphisms. The latter implies that the energy-momentum divergence tensor is equal to the gradient of a (undetermined) scalar function:

$$T^{\mu}_{\nu;\mu} = \Theta_{;\nu}, \tag{7}$$

On one hand, it is entirely permissible to set the gradient of $\Theta$ to zero. If this is conducted, we recover (2), with $\Lambda$ appearing as an integration constant. On the other hand, one can also choose,

$$\Theta = \frac{R}{4} + 2\pi G T. \tag{8}$$

As it will be seen later, if the energy-momentum tensor is not conserved ($\Theta \neq 0$), the final system of equations is underdetermined, and a supplementary ansatz must be introduced to close consistently the system. In practice, this means that the functional form of the relation (8) depends on the ansatz to be adopted. Due to this, from now on, we will identify $\Theta \equiv -\Lambda$, in order to keep in contact with the usual notation in the literature for decaying vacuum models. If $\Lambda$ is constant, as already stressed, we return to the GR equations in the presence of a cosmological constant. But, if $\Lambda$ is a function of the space-time coordinates, we end up with the following set of equations,

$$R_{\mu\nu} - \frac{1}{4}g_{\mu\nu}R = 8\pi G\left\{T_{\mu\nu} - \frac{1}{4}g_{\mu\nu}T\right\}, \tag{9}$$

$$T^{\mu}_{\nu;\mu} = -\Lambda_{;\nu}. \tag{10}$$

To proceed with a dynamical $\Lambda$, we must have some guiding physical perspectives. Equations (9) and (10) represent generally the idea of a decaying vacuum. A decaying

vacuum may be an effective mechanism to begin with a huge initial value for the vacuum energy density, reaching later a small value as observed today. To compute this decaying process, a deep comprehension of the dynamics of quantum fields in a gravitational field is required. Perhaps this may require a full quantum gravity theory. In the absence of complete quantum gravity theory, we can only try to guess, by using some reasonable assumptions, how the vacuum can decay and that is the usual approach in the literature, for example, the case of running gravitational coupling, for example, Ref. [20–22], or scale dependent gravity, see Ref. [23].

This is equivalent (up to the restriction in the diffeomorphic class of transformation) to the GR equations in the presence of a dynamical cosmological term:

$$R_{\mu\nu} - \frac{1}{2}g_{\mu\nu}R \;=\; 8\pi G T_{\mu\nu} + g_{\mu\nu}\Lambda, \tag{11}$$

$$T^{\mu}_{\nu\,;\mu} = -\Lambda_{;\nu}, \tag{12}$$

provided that $\Lambda$ is identified with $\Theta$ as given by (8). Hence, the non-conservation of the energy-momentum tensor allows us to map the UG theory into GR equipped with a dynamical cosmological term, implying an interacting-like model in the GR context.

The mapping described above seems to indicate an equivalence of UG and GR. In fact, sometimes two or more apparently different theories may describe the same phenomena (in occurrence, gravity) but in a completely equivalent way. One example, is the so-called trinity of gravity involving Riemannian geometry, torsion and non-metricity, see Refs. [24–26]. Are the relations displayed above showing that UG is GR disguised? To answer this question, we must show that the equivalence is complete. As discussed below, and also in Ref. [16], there are indications that the equivalence between UG and GR is broken at the perturbative level.

It is also convenient, for reasons that will become clear later on in the work, to write down the UG equations in a more compact form such as

$$E_{\mu\nu} = 8\pi G\,\tau_{\mu\nu}, \tag{13}$$

with the definitions,

$$E_{\mu\nu} \;=\; R_{\mu\nu} - \frac{1}{4}g_{\mu\nu}R = G_{\mu\nu} + \frac{1}{4}g_{\mu\nu}R, \tag{14}$$

$$\tau_{\mu\nu} \;=\; T_{\mu\nu} - \frac{1}{4}g_{\mu\nu}T. \tag{15}$$

We will call $E_{\mu\nu}$ the unimodular gravitational tensor and $\tau_{\mu\nu}$ the unimodular energy-momentum tensor.

## 3. Equations for a Symmetric and Static Configuration

In this section, we will write down the general expressions for a symmetric and static configuration. In the Appendix A the corresponding expressions with a time dependence will be derived, which are necessary to perform the perturbative analysis to be described later.

Let us consider the metric,

$$ds^2 = e^{2\gamma}dt^2 - e^{2\alpha}dr^2 - e^{2\beta}d\Omega^2. \tag{16}$$

The non-vanishing Christoffel symbols are the following.

$$\Gamma^0_{10} = \gamma' \quad,\quad \Gamma^1_{00} = e^{2(\gamma-\alpha)}\gamma', \tag{17}$$

$$\Gamma^1_{11} = \alpha' \quad,\quad \Gamma^1_{22} = -e^{2(\beta-\alpha)}\beta' \quad,\quad \Gamma^1_{33} = -e^{2(\beta-\alpha)}\beta'\sin^2\theta, \tag{18}$$

$$\Gamma^2_{12} = \Gamma^3_{12} = \beta', \quad \Gamma^2_{33} = -\sin\theta\cos\theta \quad,\quad \Gamma^3_{23} = \cot\theta. \tag{19}$$

Primes denote derivatives with respect to $r$.

Also, the non-vanishing components of the Ricci tensor and the Ricci scalar are the following.

$$
\begin{aligned}
R_{00} &= e^{2(\gamma-\alpha)}[\gamma'' + \gamma'(\gamma' + 2\beta' - \alpha')], & (20)\\
R_{11} &= -\gamma'' - 2\beta'' + \gamma'(\alpha' - \gamma') + 2\beta'(\alpha' - \beta'), & (21)\\
R_{22} &= 1 - e^{2(\beta-\alpha)}[\beta'' + \beta'(\gamma' + 2\beta' - \alpha')], & (22)\\
R_{33} &= R_{22}\sin^2\theta, & (23)\\
R &= -2e^{-2\beta} + 2e^{-2\alpha}[\gamma'' + 2\beta'' + 3\beta'^2 + \gamma'(\gamma' + 2\beta' - \alpha') - 2\alpha'\beta']. & (24)
\end{aligned}
$$

Consequently, the non-vanishing components of the unimodular gravitational tensor defined in (14) are the following:

$$
E_{00} = e^{2(\gamma-\alpha)}\left[\frac{\gamma''}{2} - \beta'' - \frac{3}{2}\beta'^2 + \frac{\gamma'}{2}(\gamma' + 2\beta' - \alpha') + \beta'\alpha'\right] + \frac{e^{2(\gamma-\beta)}}{2}, \tag{25}
$$

$$
E_{11} = -\frac{\gamma''}{2} - \beta'' - \frac{\beta'^2}{2} - \frac{\gamma'}{2}(\gamma' - \alpha') + \beta'(\alpha' + \gamma') - \frac{e^{2(\alpha-\beta)}}{2}, \tag{26}
$$

$$
E_{22} = \frac{1}{2} + \frac{e^{2(\beta-\alpha)}}{2}[\gamma'' - \beta'^2 + \gamma'(\gamma' - \alpha')], \tag{27}
$$

$$
E_{33} = E_{22}\sin^2\theta. \tag{28}
$$

The left-hand side of the UG field equations for the symmetric and static configuration has been set up. The next step is to characterize the source field. In the next couple of sections, the electromagnetic field and a scalar field will be considered as sources of the gravitational field.

## 4. The Electromagnetic Field

In the case of an electromagnetic field as the source of the energy-momentum tensor one has,

$$
8\pi G T_{\mu\nu}^{EM} = -2\left\{F_{\mu\rho}F_\nu^\rho - \frac{1}{4}g_{\mu\nu}F_{\rho\sigma}F^{\rho\sigma}\right\}. \tag{29}
$$

It is worth mentioning that it has zero trace:

$$
T^{EM} = 0. \tag{30}
$$

Equations (5) and (6) become,

$$
R_{\mu\nu} - \frac{1}{4}g_{\mu\nu}R = -2\left\{F_{\mu\rho}F_\nu^\rho - \frac{1}{4}g_{\mu\nu}F_{\rho\sigma}F^{\rho\sigma}\right\}, \tag{31}
$$

$$
F^{\mu\rho}{}_{;\mu}F_{\nu\rho} = -\frac{R_{;\nu}}{8}. \tag{32}
$$

Remark that, contrary to GR, the traceless character of the energy-momentum tensor does not imply $R = 0$, unless the Maxwell equations are obeyed.

Imposing the spherical symmetry, the only non-vanishing component is $F^{01} = E \equiv E(r)$. Then, the equations are:

$$
\frac{\gamma''}{2} - \beta'' - \frac{3}{2}\beta'^2 + \frac{\gamma'}{2}(\gamma' + 2\beta' - \alpha') + \beta'\alpha' + \frac{e^{2(\alpha-\beta)}}{2} = e^{2\gamma+4\alpha}E^2, \tag{33}
$$

$$
-\frac{\gamma''}{2} - \beta'' - \frac{\beta'^2}{2} + \frac{\gamma'}{2}(\alpha' + 2\beta' - \gamma') + \beta'\alpha' - \frac{e^{2(\alpha-\beta)}}{2} = -e^{2\gamma+4\alpha}E^2, \tag{34}
$$

$$
\frac{1}{2}\left[\gamma'' - \beta'^2 + \gamma'(\gamma' - \alpha')\right] + \frac{e^{2(\alpha-\beta)}}{2} = e^{2\gamma+4\alpha}E^2, \tag{35}
$$

$$
(E^2)' + 2(\alpha' + \gamma' + 2\beta')E^2 = \frac{e^{-2(\alpha+\gamma)}}{4}R', \tag{36}
$$

with $R$ given by (24).

Until now, no coordinate condition has been imposed. Adding (33) and (34), we obtain,

$$\beta'(\alpha' + \gamma') - \beta'' - \beta'^2 = 0. \tag{37}$$

The use of the quasi-global coordinates, with $\alpha = -\gamma$, leads to,

$$\beta = \log r. \tag{38}$$

As in the usual Reissner–Nordström (RN) solution in GR, there is a center at $r = 0$. Equation (8), with $T = 0$ and with the identification of $\Theta$ with $-\Lambda$ implies in $R = -4\Lambda$. The equations of motion reduce to,

$$\gamma'' + 2\gamma'^2 - \frac{1}{r^2} + \frac{e^{-2\gamma}}{r^2} = 2e^{-2\gamma}E^2, \tag{39}$$

$$(E^2)' + 4\frac{E^2}{r} = -\Lambda', \tag{40}$$

In order to proceed further, we must impose a condition. This is a crucial step in working with UG as already stressed in [16]. One possibility is to fix the $R = -4\Lambda \equiv$ constant. This leads to the Reisnner–Nordström–de Sitter (RNdS) solution. In fact, this implies to recover the conservation law $F^{\mu\nu}{}_{;\mu} = 0$. If $\Lambda = 0$, we re-obtain the RN solution. If $\Lambda > 0$, the RNdS solution is obtained, and if $\Lambda < 0$, the Reisnner–Nordström-(Anti) de Sitter (RNAdS) solution is recovered, as it will be seen below. On the other hand, there are also also other possibilities that are to be explored since $\Lambda$ can be nonconstant, covering the possibility of a dynamical cosmological term.

Three cases will be considered, namely a constant and two dynamical cosmological terms, corresponding to either the usual or the modified conservation laws.

*4.1. Constant Cosmological Term*

If $\Lambda = $ constant,

$$E = \frac{Q}{r^2}, \tag{41}$$

after identifying an integration constant with the total charge $Q$. The Coulomb law is recovered, as in the RN solution.

Using the quasi-global coordinate condition and the solution for the electric field $E$, Equation (33) becomes:

$$e^{2\gamma}(\gamma'' + 2\gamma'^2) - \frac{e^{2\gamma}}{r^2} = \frac{1}{r^2} + 2\frac{Q^2}{r^4}. \tag{42}$$

Defining $A = e^{2\gamma}$, the equation takes the form,

$$A'' - 2\frac{A}{r^2} = \frac{2}{r^2} + 4\frac{Q^2}{r^4} \tag{43}$$

This is a second-order, linear, non-homogeneous differential equation whose solution is

$$A = 1 + \frac{C_1}{r} + \frac{Q^2}{r^2} + \frac{C_2}{3}r^2, \tag{44}$$

$C_{1,2}$ being integration constants. Inserting this solution into the condition $R = -\Lambda$, it is satisfied provided $C_2 = -\Lambda$, while $C_1$ remains arbitrary, being fixed by using the Newtonian limit.

The final solution is given by,

$$A = 1 - 2\frac{GM}{r} + \frac{Q^2}{r^2} - \frac{\Lambda}{3}r^2. \tag{45}$$

This is the RNdS solution. It coincides with the static and spherically symmetric solution in GR with an electromagnetic field and a cosmological constant. This could be expected from the beginning since UG (satisfying the usual conservation laws) leads to the same field equations as GR with a cosmological term, with the only (but important, as we will see later) difference that UG is restricted to transverse diffeomorphism instead of the full diffeomorphism.

### 4.2. Varying Cosmological Term

For a varying cosmological term, it is necessary to impose an ansatz on the behavior of the function $\Lambda$. This is also true in GR when the cosmological term is dynamical. Since a static and spherically symmetric configuration is considered, the cosmological term must be a function on the coordinate $r$ only: $\Lambda \equiv \Lambda(r)$.

Let us restrict ourselves again to the condition $R = -4\Lambda$. Using the previous results and also identifying $\beta = \ln r$, $\alpha = -\gamma$ and $A = e^{2\gamma}$, then:

$$A'' + 4\frac{A'}{r} + 2\frac{A}{r^2} = \frac{2}{r^2} - 4\Lambda(r). \tag{46}$$

The solution for the homogenous equation is,

$$A_h = \frac{C_1}{r} + \frac{C_2}{r^2}. \tag{47}$$

To obtain the inhomogeneous solution, we write,

$$A = \frac{f}{r^2}, \tag{48}$$

obtaining,

$$f'' = 2 - 4r^2\Lambda(r). \tag{49}$$

with a solution that depends on $r$:

$$f = r^2 - 4\int\left[\int^r r'^2\Lambda(r')dr'\right]dr. \tag{50}$$

We will consider two different configurations for the function $\Lambda(r)$, corresponding to two distinct behaviors both asymptotically as well as at the center ($r = 0$).

#### 4.2.1. Case A

First, it imposes a power law behavior for $\Lambda(r)$,

$$\Lambda(r) = \Lambda_0 + \Lambda_1 r^p, \tag{51}$$

with $\Lambda_{0,1}$ constants.

The final solution is given by the following expressions.

- $p \neq -4$:

$$A = 1 - \frac{2GM}{r} + \frac{Q^2}{r^2} - \frac{\Lambda_0}{3}r^2 - \frac{4\Lambda_1}{(p+3)(p+4)}r^{p+2}, \tag{52}$$

$$E^2 = \frac{Q^2}{r^4} - \frac{p}{p+4}\Lambda_1 r^p; \tag{53}$$

- $p = -4$:

$$A = 1 - \frac{2GM}{r} + \frac{Q^2}{r^2} - \frac{\Lambda_0}{3}r^2 - \frac{16\Lambda_1}{9r}\left\{3(\ln r)^2 + \ln r\right\}, \tag{54}$$

$$E^2 = \frac{Q^2}{r^4} + 4\frac{\Lambda_1}{r^4}\ln r. \tag{55}$$

The case $p = -4$ is clearly pathological since the electric field becomes imaginary near $r = 0$ when $\Lambda_1 > 0$ or for large $r$ if $\Lambda_1 < 0$. For $p \neq -4$ a change of sign of $E^2$ can be avoided by choosing $\Lambda_1 > 0$ for $-4 < p < 0$, or $\Lambda_1 < 0$ for $p < -4$ or $p > 0$. The values $p = 0, -3$ correspond to the cases already included in the constants $C_1$ and $C_2$ of the homogenous solution.

The solution $p \neq 0$, with the required conditions to avoid an imaginary electric field, contains either multiple horizon black holes, with a singularity at $r = 0$, or naked singularities similar to the de Sitter–Reissner–Nordström (dSRN) solution in GR but the metric functions, in the UG case, may present a different shape but without change the general properties. These solutions are asymptotically non-flat except if $\Lambda_0 = 0$ and $p > -2$. The corresponding equations in GR equipped with a cosmological term with the same functional dependence, using the same symmetries, lead to the same solution as it can be explicitly verified.

### 4.2.2. Case B

We will exploit now the functional form,

$$\Lambda(r) = \Lambda_0 + \frac{\Lambda_1}{(r^2 + a^2)^2}. \tag{56}$$

If $\Lambda_0 = 0$, this functional form represents an asymptotically constant cosmological term near the origin, which becomes zero at infinity. Following the same steps as the previous case, the final form of the metric function is:

$$
\begin{aligned}
A &= 1 - \frac{2GM}{r} + \frac{Q^2}{r^2} - \frac{\Lambda_0}{3} r^2 \\
&\quad - 2\frac{\Lambda_1}{r^2}\left[\frac{r}{a}\arctan\frac{r}{a} - \ln\left(1 + \frac{r^2}{a^2}\right)\right], \tag{57}
\end{aligned}
$$

$$
E^2 = \frac{Q^2}{r^4} + \Lambda_1\left\{-\frac{1}{(r^2 + a^2)^2} + 2\left[\frac{1}{r^2 a^2} - \frac{1}{r^4}\ln\left(1 + \frac{r^2}{a^2}\right)\right]\right\}. \tag{58}
$$

Again, the same solution is obtained in the GR with a varying cosmological term given by (56). There are multiple horizons and naked singularities, as in the previous case. No change of sign in the $E^2$ term can be assured by imposing $\Lambda_1 > 0$.

In all the cases discussed above, the presence of the cosmological term, constant or not, introduces new features in the solutions with respect to the usual Reissner–Nordström (RN) solution but does not remove the singularity at $r = 0$. The behavior of this class of solution is depicted in Figure 1 for some parameter values.

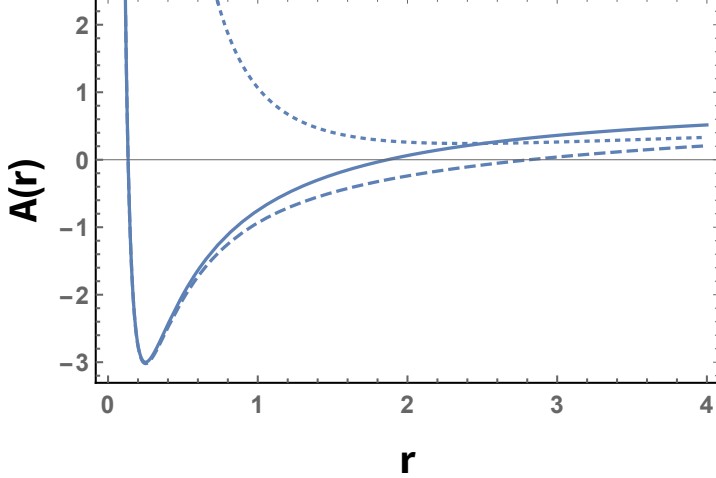

**Figure 1.** Solution of the case B using $\Lambda_0 = 0$, $\Lambda_1 = 1/2$, with $m = 1$, $Q = 0.5$ (Dashed), and $Q = 1.5$ (Dotted). The continuous line is the RN solution with $\Lambda_0 = 0$, $\Lambda_1 = 0$, with $m = 1$ and $Q = 0.5$.

## 5. Scalar Field

The energy-momentum tensor for a self-interacting scalar field is,

$$T_{\mu\nu} = \epsilon\left(\phi_{;\mu}\phi_{;\nu} - \frac{1}{2}g_{\mu\nu}\phi_{;\rho}\phi^{;\rho}\right) + g_{\mu\nu}V(\phi). \tag{59}$$

The ordinary scalar field is denoted by $\epsilon = +1$ and the phantom scalar field by $\epsilon = -1$. In GR, in four dimensions, black holes exist only for the phantom case [27,28].

Inserting the expression for the energy-momentum tensor (59) in the UG equations one obtains,

$$R_{\mu\nu} - \frac{1}{4}g_{\mu\nu}R = \epsilon\left(\phi_{;\mu}\phi_{;\nu} - \frac{1}{4}g_{\mu\nu}\phi_{;\rho}\phi^{;\rho}\right), \tag{60}$$

$$\frac{R_{;\nu}}{4} = \epsilon\left(\phi_{;\nu}\Box\phi + \frac{\phi^{;\rho}\phi_{;\nu;\rho}}{2}\right). \tag{61}$$

One distinguishing feature of the above equations is the absence of the potential (or, as before, the cosmological term, which is the particular case of a constant potential): it naturally disappears due to the traceless structure of the UG equations. Equation (61) can be written as,

$$\left(\frac{R}{4} - \frac{\epsilon}{4}\phi_{\rho}\phi^{;\rho}\right)_{;\nu} = \epsilon\phi_{;\nu}\Box\phi. \tag{62}$$

Identifying,

$$\frac{R}{4} - \frac{\epsilon}{4}\phi_{\rho}\phi^{;\rho} = -V(\phi), \tag{63}$$

Equations (60) and (61) take the following form,

$$R_{\mu\nu} - \frac{1}{2}g_{\mu\nu}R = \epsilon\left(\phi_{;\mu}\phi_{;\nu} - \frac{1}{2}g_{\mu\nu}\phi_{;\rho}\phi^{;\rho}\right) + g_{\mu\nu}V(\phi), \tag{64}$$

$$\Box\phi = -\epsilon V_{\phi}(\phi). \tag{65}$$

In this way, we recover the GR equations equipped with a self-interacting scalar field.

In a static and spherically symmetric configuration, the UG field Equation (60) read,

$$\frac{\gamma''}{2} - \beta'' - \frac{3}{2}\beta'^2 + \frac{\gamma'}{2}(\gamma' + 2\beta' - \alpha') + \beta'\alpha' + \frac{e^{2(\alpha-\beta)}}{2} = \epsilon\frac{\phi'^2}{4}, \tag{66}$$

$$-\frac{\gamma''}{2} - \beta'' - \frac{\beta'^2}{2} - \frac{\gamma'}{2}(\gamma' - \alpha') + \beta'(\alpha' + \gamma') - \frac{e^{2(\alpha-\beta)}}{2} = \epsilon\frac{3}{4}\phi'^2, \tag{67}$$

$$\frac{1}{2}[\gamma'' - \beta'^2 + \gamma'(\gamma' - \alpha')] + \frac{e^{2(\alpha-\beta)}}{2} = -\epsilon\frac{\phi'^2}{4}. \tag{68}$$

Combining these equations, we have the following relations:

$$\gamma'' - \beta'' - 2\beta'^2 + \gamma'(\gamma' + \beta' - \alpha') + \alpha'\beta' + e^{2(\alpha-\beta)} = 0, \tag{69}$$

$$-\beta'' - \beta' + \beta'(\gamma' + \alpha') = \frac{\phi'^2}{2}. \tag{70}$$

Remark that in the UG equations, there is no potential, even if it appears in the energy-momentum tensor. Moreover, there are three metric functions (which can be reduced to two functions by gauging the radial coordinate) and the scalar field to be determined, and just two independent equations, (69) and (70). Hence an ansatz must be introduced. From the conservation law, we have the relation,

$$R + e^{-\alpha}\phi'^2 = -4V(\phi), \tag{71}$$

where $V(\phi)$ is a function to be determined. We have slightly changed the notation ($V$ instead of $\Lambda$) to identify the unknown function with the potential. With this identification, the UG equations become identical to the GR equations with a potential. In GR, the potential must be chosen. In UG a functional form for the scalar field (or for one of the metric functions) must be chosen. There is a correspondence between the choice of the functional form of the scalar field and the choice of the potential in GR.

Two possible examples are the following.

1.  If the scalar field is chosen such that,

$$\phi = -\epsilon\frac{C}{2k}P, \tag{72}$$

$$P = 1 - 2\frac{k}{\rho}, \tag{73}$$

we find,

$$ds^2 = P^a dt^2 - P^{-a}d\rho^2 - P^{1-a}\rho^2 d\Omega^2, \tag{74}$$

$$a^2 = 1 - \epsilon\frac{C^2}{k^2}. \tag{75}$$

Here, $\rho$ denotes the radial coordinate in the quasi-global coordinate system.

Using (71) we find $V = 0$. This solution represents a black hole only if $\epsilon = -1$. This solution has been determined in the GR context in Ref. [29].

2.  The regular black hole determined in Ref. [28], is also the solution in the UG case, without a potential. Imposing that the scalar field is given by,

$$\psi = \frac{\phi}{\sqrt{2}} = \arctan\frac{\rho}{b}, \tag{76}$$

the metric is then given, in the quasi-global coordinates, by,

$$ds^2 = Adt^2 - \frac{d\rho^2}{A} - r^2(\rho)d\Omega^2, \tag{77}$$

$$A = 1 + \frac{c}{b^2}r^2 + \frac{\rho_0}{b^3}\left(b\rho + r^2\arctan\frac{\rho}{b}\right). \tag{78}$$

In these expressions $b$, $c$ and $\rho_0$ are constants. Using relation (71) the potential in GR context is given by,

$$V - \frac{c}{b^3}\left(3 - 2\cos^2\psi\right) - \frac{\rho_0}{b^3}\left[3\sin\psi\cos\psi + \psi\left(3 - 2\cos^2\psi\right)\right], \tag{79}$$

the same used in the GR context.

## 6. Remarks on the Birkhoff Theorem and the Stability of the Solutions

Initially, we present a straightforward argument to demonstrate that, in the cases of both electromagnetism and scalar fields, the Birkhoff theorem holds the same significance as it does in GR. For electric-charged static solutions in GR, the Birkhoff theorem is valid. The same occurs for the corresponding solution in the UG. This can be seen by supposing radial time-dependent configurations. The argumentation follows the same reasoning used in GR, see for example [30]. From the expression presented in the Appendix, and considering only a radial electric field, the $0 - 1$ component of the field equations, using the Schwarzschild coordinate system, with $\beta = \ln r$, implies that $\alpha$ must be time-independent since the right-hand side of the equation is zero for a pure radial electric field. Combining

equations $0 - 0$ and $1 - 1$, it comes out that $\alpha = -\gamma$. Hence, all metric functions are time-independent.

For the scalar field case, the Birkhoff theorem is not valid because the right-hand side contains a term of the time $\dot{\phi}\phi'$ which forbids considering the metric function $\alpha$ as time independent, as it happens in the GR case. The Birkhoff theorem is verified only if the scalar field is static [31].

In the two examples discussed in the previous section, having the scalar field as the source of the geometry, and considering the GR context, the solutions are unstable, except for the regular solution in the very special case where the minimum of the areal function coincides with the horizon [32,33]. However, this result can change in the UG context since the unimodular condition implies new relations for the perturbed functions that are absent in GR.

We will illustrate the special features of the perturbative analysis considering the case of black holes with a scalar field. Only radial perturbations will be considered. In the GR context, this is enough to conclude about the instability of the solution [32]. We will show that in the UG, if we try to follow the same procedure as in GR, the perturbations at first order are strictly zero due to the unimodular condition.

The unimodular condition implies,

$$g = \det g_{\mu\nu} = e^{\alpha+\gamma+2\beta} = \xi. \tag{80}$$

Since the function $\xi$ is fixed, the unimodular condition leads, at linear perturbative order,

$$\delta\alpha + \delta\gamma + 2\delta\beta = 0. \tag{81}$$

There is still the freedom to impose a coordinate condition due to the diffeomorphic (even if transverse) invariance. The choice $\delta\beta = 0$ is related to the gauge-invariant variables [32]. In fact, in GR this choice is equivalent to the full employment of gauge invariant variables and we will suppose that this property is also valid in UG. Hence, we end up with the conditions,

$$\delta\alpha = -\delta\gamma, \quad \delta\beta = 0. \tag{82}$$

We write down the perturbations in a generic way as

$$\delta f(x,t) = f(x)e^{-i\omega t}. \tag{83}$$

The perturbed equations, under the conditions above, are the following:

$$\delta\gamma'' + 4\gamma'\delta\gamma' - \left\{\omega^2.e^{-4\gamma} + 2e^{-2(\gamma+\beta)}\right\}\delta\gamma = \phi'\delta\phi', \tag{84}$$

$$\delta\gamma'' + 4\gamma'\delta\gamma' - \left\{\omega^2.e^{-4\gamma} + 2e^{-2(\gamma+\beta)}\right\}\delta\gamma = -3\phi'\delta\phi', \tag{85}$$

$$\delta\gamma'' + 4\gamma'\delta\gamma' - \left\{\omega^2.e^{-4\gamma} + 2e^{-2(\gamma+\beta)}\right\}\delta\gamma = -\phi'\delta\phi', \tag{86}$$

$$-2\beta'\delta\gamma = \phi'\delta\phi. \tag{87}$$

It is clear that the equations are consistent only in the trivial case: $\delta\phi = \delta\gamma = 0$. Hence, it is not possible to obtain information on the stability of the solution, at least at the linear level and follow a procedure close to that used in GR. This indicates a distinguishing feature of unimodular gravity in comparison with GR.

However, some cautions are necessary. In cosmology, the Newtonian gauge is equivalent to the gauge-invariant formalism in GR, in the sense that, at least in the absence of anisotropic stress, the final equations are the same [34]. In Ref. [17] it has been argued that, in cosmology and in the UG context, the gauge-invariant formalism can be applied while the Newtonian gauge is not. Does the same happen here, when static, spherically

symmetric perturbations are considered? In our point of view, this asks for a separate analysis, especially if we take into account the point of view presented in Ref. [35]. We must remember, on the other hand, that the same solution may appear in different theories, but with different perturbative behavior, see an example in Ref. [36] where the same wormhole solution has been analyzed at background and perturbative levels in three different theoretical contexts.

## 7. Conclusions

Unimodular gravity (UG) is one of the first alternatives to General Relativity (GR). It is a geometric theory that is invariant with respect to a restricted diffeomorphic class of transformations, the transverse diffeomorphism, due to the imposition of a constraint on the determinant of the metric. In UG the usual conservation of the energy-momentum tensor is not assured: The conservation of the energy-momentum tensor is a choice. If it is imposed, UG becomes in principle equivalent to GR with a cosmological term. However, the restriction on the determinant of the metric may lead to some important new features at the perturbative level. We have shown here that if the conservation of the energy-moment tensor is relaxed, UG becomes equivalent to GR with a dynamical cosmological term, with still the same important difference due to the UG constraint which can manifested at the perturbative level.

We have discussed, in this context, the static and spherically symmetric solutions in UG. For the vacuum configuration, the Schwarzschild solution is also verified in UG. The same occurs with the Reissner–Nordström solution, but only if the energy-momentum tensor is conserved. If not, the dynamical cosmological term induces new features, but it does not prevent the appearance of the singularity at $r = 0$. Similar features appear in the case when a scalar field appears as the main source. In this case, the potential term, representing the self-interaction of the scalar field, disappears in the UG context and an ansatz must be imposed in order to close the set of equations. This mounts, in the GR context, to choose a given potential for the scalar field. For a discussion of the UG in static, spherical configurations but focusing on compact objects, see Ref. [37].

We have shown that the Birkhoff theorem follows the same features as in GR, being satisfied for a charged solution, being possibly violated for a dynamical scalar field. The linear radial perturbations have been analyzed when a scalar field is present. Once more, GR black hole solutions are generically unstable in the latter case. In UG, using the gauge invariant approach employed in GR and restricting to radial perturbations, the condition on the determinant of the metric leads to vanishing perturbations at linear order, and possibly also for higher order. As already discussed in the cosmological context, this result seems to point out a breaking of the equivalence of UG and GR at the perturbative level. There are other viewpoints on the implementation of the UG constraints in performing a perturbative analysis, see for example Ref. [35]. However, the results reported here indicate that a direct application of the procedures used in GR combined with the unimodular constraint may lead to conclusions different from those obtained in GR.

In fact, the main open problem related to the results obtained above refers to the perturbative analysis. As pointed out in previous works (see Ref. [16] and references therein) the main difference between GR and UG is related to the perturbative analysis. While the UG gravity can be mapped into GR with a cosmological constant, when the energy-momentum tensor is conserved, or in a dynamical vacuum model, when the energy-momentum tensor is not conserved, previous investigations indicate that such mapping seems not to be complete when perturbations are considered even at the perturbative level. We have given an example here, applying directly the relations coming from the unimodular constraints besides the usual coordinate condition. However, there are claims (for example, [35]) that this issue is not closed and, for example, all coordinate conditions can be used in UG in spite of the unimodular constraint in the determinant of the metric. We intend to present in the future a more concrete and detailed analysis of this problem.

**Author Contributions:** Conceptualization, J.C.F. and H.V.; methodology, J.C.F., H.V. and M.H.D.; validation, M.H.D.; investigation, J.C.F., H.V. and M.H.D.; writing—original draft preparation, J.C.F., H.V. and M.H.D. All authors have read and agreed to the published version of the manuscript.

**Funding:** We thank CNPq, FAPES and FAPEMIG for partial financial support.

**Data Availability Statement:** Data are contained within the article.

**Acknowledgments:** We thank K.A. Bronnikov for enlighting discussions on some aspects of the problem treated in this work and L.F. de Oliveira Guimarães for his remarks on the text.

**Conflicts of Interest:** The authors declare no conflict of interest.

## Appendix A. The Spherically Symmetric Non Static Metric

A dynamical spherically symmetric metric, admitting radial oscillations, is given by,

$$ds^2 = e^{2\gamma(t,r)}dt^2 - e^{2\alpha(t,r)}dr^2 - e^{2\beta(t,r)}d\Omega^2. \tag{A1}$$

The non-vanishing Christoffel symbols are the following.

$$\Gamma^0_{00} = \dot{\gamma}, \quad \Gamma^0_{10} = \gamma', \quad \Gamma^0_{11} = e^{2(\alpha-\gamma)}\dot{\alpha}, \tag{A2}$$

$$\Gamma^0_{22} = e^{2(\beta-\gamma)}\dot{\beta}, \quad \Gamma^0_{33} = \Gamma^0_{22}\sin^2\theta, \tag{A3}$$

$$\Gamma^1_{00} = e^{2(\gamma-\alpha)}\gamma', \quad \Gamma^1_{01} = \dot{\alpha}, \quad \Gamma^1_{11} = \alpha', \tag{A4}$$

$$\Gamma^1_{22} = -e^{2(\beta-\alpha)}\beta', \quad \Gamma^1_{33} = \Gamma^1_{22}\sin^2\theta, \tag{A5}$$

$$\Gamma^2_{02} = \Gamma^3_{03} = \dot{\beta}, \quad \Gamma^2_{12} = \Gamma^3_{12} = \beta', \tag{A6}$$

$$\Gamma^2_{33} = -\sin\theta\cos\theta \quad, \quad \Gamma^3_{23} = \cot\theta. \tag{A7}$$

The dots mean derivatives with respect to $t$ and the primes with respect to $r$.

The non-vanishing components of the Ricci tensor and the Ricci scalar are the following.

$$\begin{aligned} R_{00} &= -\ddot{\alpha} - 2\ddot{\beta} + \dot{\gamma}(\dot{\alpha} + 2\dot{\beta}) - \dot{\alpha}^2 - 2\dot{\beta}^2 \\ &+ e^{2(\gamma-\alpha)}[\gamma'' + \gamma'(\gamma' + 2\beta' - \alpha')], \end{aligned} \tag{A8}$$

$$\begin{aligned} R_{11} &= e^{2(\alpha-\gamma)}\left\{\ddot{\alpha} + \dot{\alpha}(\dot{\alpha} - \dot{\gamma} + 2\dot{\beta})\right\} \\ &- \gamma'' - 2\beta'' + \gamma'(\alpha' - \gamma') + 2\beta'(\alpha' - \beta'), \end{aligned} \tag{A9}$$

$$\begin{aligned} R_{22} &= 1 + e^{2(\beta-\gamma)}[\ddot{\beta} + \dot{\beta}(\dot{\alpha} + 2\dot{\beta} - \dot{\gamma})] \\ &- e^{2(\beta-\alpha)}[\beta'' + \beta'(\gamma' + 2\beta' - \alpha')], \end{aligned} \tag{A10}$$

$$R_{33} = R_{22}\sin^2\theta, \tag{A11}$$

$$R_{01} = 2\left\{\dot{\beta}' + \dot{\beta}(\gamma' - \beta') + \dot{\alpha}\beta'\right\}, \tag{A12}$$

$$\begin{aligned} R &= -2e^{-2\beta} + 2e^{-2\alpha}[\gamma'' + 2\beta'' + 3\beta'^2 + \gamma'(\gamma' + 2\beta' - \alpha') - 2\alpha'\beta'] \\ &- 2e^{-2\gamma}[\ddot{\alpha} + 2\ddot{\beta} + 3\dot{\beta}^2 + \dot{\alpha}(\dot{\alpha} + 2\dot{\beta} - \dot{\gamma}) - 2\dot{\gamma}\dot{\beta}] \end{aligned} \tag{A13}$$

The non-vanishing components of the unimodular gravitational tensor,

$$E_{\mu\nu} = R_{\mu\nu} - \frac{1}{4}g_{\mu\nu}R, \tag{A14}$$

are the following:

$$
\begin{aligned}
E_{00} &= -\frac{\ddot{\alpha}}{2} - \ddot{\beta} + \frac{\dot{\gamma}}{2}(\dot{\alpha} + 2\dot{\beta}) - \frac{\dot{\alpha}^2}{2} - \frac{\dot{\beta}^2}{2} + \dot{\alpha}\dot{\beta} \\
&\quad + e^{2(\gamma-\alpha)}\left[\frac{\gamma''}{2} - \beta'' - \frac{3}{2}\beta'^2 + \frac{\gamma'}{2}(\gamma' + 2\beta' - \alpha') + \beta'\alpha'\right] + \frac{e^{2(\gamma-\beta)}}{2}, \quad \text{(A15)}
\end{aligned}
$$

$$
\begin{aligned}
E_{11} &= e^{2(\alpha-\gamma)}\left[\frac{\ddot{\alpha}}{2} - \ddot{\beta} - \frac{3}{2}\dot{\beta}^2 + \frac{\dot{\alpha}}{2}(\dot{\alpha} + 2\dot{\beta} - \dot{\gamma}) + \dot{\gamma}\dot{\beta}\right] \\
&\quad - \frac{\gamma''}{2} - \beta'' - \frac{\beta'^2}{2} - \frac{\gamma'}{2}(\gamma' - \alpha') + \beta'(\alpha' + \gamma') - \frac{e^{2(\alpha-\beta)}}{2}, \quad \text{(A16)}
\end{aligned}
$$

$$
\begin{aligned}
E_{22} &= \frac{1}{2} - \frac{e^{2(\beta-\gamma)}}{2}[\ddot{\alpha} - \dot{\beta}^2 + \dot{\alpha}(\dot{\alpha} - \dot{\gamma})] \\
&\quad + \frac{e^{2(\beta-\alpha)}}{2}[\gamma'' - \beta'^2 + \gamma'(\gamma' - \alpha')], \quad \text{(A17)}
\end{aligned}
$$

$$
E_{01} = 2[\dot{\beta}' + \dot{\beta}(\gamma' - \beta') + \dot{\alpha}\beta'], \quad \text{(A18)}
$$

$$
E_{33} = E_{22}\sin^2\theta. \quad \text{(A19)}
$$

For the static case, the above expressions reduce to,

$$
E_{00} = e^{2(\gamma-\alpha)}\left[\frac{\gamma''}{2} - \beta'' - \frac{3}{2}\beta'^2 + \frac{\gamma'}{2}(\gamma' + 2\beta' - \alpha') + \beta'\alpha'\right] + \frac{e^{2(\gamma-\beta)}}{2}, \quad \text{(A20)}
$$

$$
E_{11} = -\frac{\gamma''}{2} - \beta'' - \frac{\beta'^2}{2} - \frac{\gamma'}{2}(\gamma' - \alpha') + \beta'(\alpha' + \gamma') - \frac{e^{2(\alpha-\beta)}}{2}, \quad \text{(A21)}
$$

$$
E_{22} = \frac{1}{2} + \frac{e^{2(\beta-\alpha)}}{2}[\gamma'' - \beta'^2 + \gamma'(\gamma' - \alpha')], \quad \text{(A22)}
$$

$$
E_{33} = E_{22}\sin^2\theta. \quad \text{(A23)}
$$

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
