# Peer review of "Spherically Symmetric Configurations in Unimodular Gravity"

_universe, doi:10.3390/universe9120515_

Round 1
Reviewer 1 Report
Comments and Suggestions for Authors
The manuscript investigates some static and spherically symmetric spacetimes in the framework of unimodular gravity. By assuming the conservation or the non-conservation of the energy-momentum tensor, the solutions that recover GR and those that have a varying cosmological constant can be obtained. The authors claim that the unimodular constraint would break the equivalence between GR and UG at the perturbation level.
I have some comments:
1. In Eq. (16), the radial coordinates should be r. The authors should also mention clearly that the primes denote the derivatives with respect to r after Eqs. (17)-(19).
2. In Eq. (74), the typos d\rho^4 and d\Omega should be fixed.
3. It is well-known that the unimodular constraint g=\xi is a coordinate-dependent entity. At the background level, one can easily rewrite the metric in a form that satisfies the unimodular constraint. Therefore, using the unimodular constraint to argue that the linear perturbations vanish seems not a coordinate-independent argument to me. More explicitly, when writing down Eq. (81), a particular coordinate has been chosen. A more careful analysis of the linear perturbations for gauge invariant quantities is necessary.
Comments on the Quality of English LanguageIn some places, the authors write RG instead of GR. I believe they are typos and should be fixed.
Reviewer 2 Report
Comments and Suggestions for Authors
Dear Authors,
I have read with great interest your paper entitled “Spherically Symmetric Configurations in Unimodular Gravity”. The work contains new and original results. The topic is vey important from a theoretical and fundamental physics perspectives for a series of implications. The work is written in a very good English, although I have found some typos and minor issues (e.g., RG instead of GR, missing punctuations, words not spelled correctly. I invite the authors to carefully re-read the paper and improving these aspects.
Regarding the scientific matter, I have some minor issues to ask. If the authors reply correctly to them, I will recommend the paper for publication.
Introduction
- From line 13-26, there are several sentences without a reference. Although some aspects are known, I would suggest you to quote some papers.
- When you mention “transverse dffemorphism” besides to cite a work, give also in footnote or in the text a definition, because it is missing in the work. It is an important aspect of the topic.
General
- An important aspect to underline is the concept of equivalent gravities. As you know there are theories of gravitas that although they are formulated differently they are the same, because basically they have the same Lagrangian or equivalently the same field equations. For examples there is the so called geometric trinity of gravity, which are equivalent formulations of general relativity. You can find several papers in the literature. Can you discuss a little bit more on this aspect? Can UG be generally set equal to GR or not? Make some comparisons with the literature on such argument.
- It is not defined that the prime stays for the derivative of a function with respect to the radius. It must be added in the text.
- Please clarify the undefined acronyms, like dSR, RN
- The future implications of your work in the conclusions must be stregthened. I suggest you to add more information about future implications and perspectives of your work, in order to highlight the importance of your results.
The English is good, some typos and minor issues must be corrected.
Reviewer 3 Report
Comments and Suggestions for Authors
The present paper investigates Unimodular gravity (UG) and its potential
equivalence to General Relativity (GR). In particular, the conditions for the equivalence between the two formulations are investigated, by applying the UG to the static and spherically symmetric configurations for two concrete cases: i) the energy-momentum tensor sourced by a scalar field or ii) by the electromagnetic field. It was argued that the equivalence may be broken when analyzing the stability of the solutions at the perturbative level.
This short paper is, in my opinion, quite acceptable. On the one hand, I think the content could be better, but such a statement is subjective, so I'll focus on the objective aspects to improve the present work. My criticisms are summarized below:
- Unimodular gravity is weakly motivated. Since this is the main idea, I invite the authors to add additional paragraphs to make the paper self-contained. While it is true that the idea is well known, it might be better to revisit the logic and ingredients behind UG.
- Provide additional details on why we can assume $R = 4 \Lambda =
\text{constant}$. - If you accept a varying cosmological coupling function, the authors are forced to assume a concrete form of $\Lambda$. I understand that this is necessary to make progress, but such a necessity may mean that some physical conditions have been ignored. Could the authors explain that?
- In the same line, I suggest the authors mention some work in the context of
running vacuum models, where the cosmological constant is able to vary.
e-Print: 2308.13349 [gr-qc]
e-Print: 2307.13130 [gr-qc]
- Another similar approach where the couplings evolve is scale-dependent gravity. Such ideas have been applied to cosmology, black holes, and stars. See also
e-Print: 1801.03248 [hep-th]
e-Print: 1806.03024 [hep-th]
and references therein.
- I strongly suggest that the authors include some graphics, for educational
purposes.
Thus, given my comments, becomes clear that the paper can be improved with minimal/decent work, so, I cannot accept the paper in its current form. After the authors include my comments, I may reconsider.
Round 2
Reviewer 1 Report
Comments and Suggestions for Authors
The authors have revised the manuscript properly and I recommend the publication.
Reviewer 3 Report
Comments and Suggestions for Authors
Some changes have been made. I think this idea has more potential, but the current form of the draft is acceptable and can be published now.